# Binational cohort study comparing the management and outcomes of pregnant women with a BMI >50–59.9 kg/m² and those with a BMI ≥60 kg/m²

Stephen J McCall,[1] Zhuoyang Li,[2] Jennifer J Kurinczuk,[1] Elizabeth Sullivan,[2] Marian Knight[1]

ES and MK are joint senior authors

¹National Perinatal Epidemiology Unit, Nuffield Department of Population Health, University of Oxford, Oxford, UK
²The Australian Centre for Public and Population Health Research, University of Technology Sydney, Sydney, Australia

**Correspondence to**
Stephen J McCall;
stephen.mccall@npeu.ox.ac.uk

## ABSTRACT

**Objectives** To compare the management, maternal and perinatal outcomes of women with a body mass index (BMI) ≥60 kg/m² with women with a BMI >50–59.9 kg/m².

**Design** International collaborative cohort study.

**Setting** Binational study in the UK and Australia.

**Participants** UK: all pregnant women, and Australia: women who gave birth (birth weight ≥400 g or gestation ≥20 weeks)

**Methods** Data from the Australasian Maternity Outcomes Surveillance System and UK Obstetric Surveillance System. Management, maternal and infant outcomes were compared between women with a BMI ≥60 kg/m² and women with a BMI >50–59.9 kg/m², using unconditional logistic regression.

**Results** The sociodemographic characteristics and previous medical histories were similar between the 111 women with a BMI ≥60 kg/m² and the 821 women with a BMI >50–59.9 kg/m². Women with a BMI ≥60 kg/m² had higher odds of thromboprophylaxis usage in both the antenatal (24% vs. 12%; OR 2.25, 95% CI 1.39 to 3.64) and postpartum periods (78% vs. 66%; OR 1.68, 95% CI 1.04 to 2.70). Women with BMI ≥60 kg/m² had nearly double the odds of pre-eclampsia/eclampsia (adjusted OR 1.83 (95% CI 1.01 to 3.30)). No other maternal or perinatal outcomes were statistically significantly different. Severe adverse outcomes such as perinatal death were uncommon in both groups thus limiting the power of these comparisons. The rate of perinatal deaths was 18 per 1000 births for those with BMI ≥60 kg/m²; 12 per 1000 births for those with BMI >50–59.9 kg/m²; those with BMI ≥60 kg/m² had a non-significant increased odds of perinatal death (unadjusted OR 1.46, 95% CI 0.31 to 6.74).

**Conclusions** Women are managed differently on the basis of BMI even at this extreme as shown by thromboprophylaxis. The pre-eclampsia result suggests that future research should examine whether weight reduction of any amount prior to pregnancy could reduce poor outcomes even if women remain extremely obese.

## INTRODUCTION

Obesity is a major risk factor for non-communicable disease and morbidity in later life. It has reached epidemic levels in many

## Strengths and limitations of this study

► Population-based study examining extreme obesity using national data from the UK and Australia.
► International collaborative studies allow the examination of rare exposures.
► This study lacked the power to examine many maternal and perinatal outcomes despite having data from two national studies
► Some outcomes were not comparable between Australia and the UK and so could not be explored.

high-income settings across all age ranges. Obesity is defined as a body mass index (BMI) of ≥30 kg/m². Increasing rates of obesity in the general population are associated with an increasing trend towards obesity in pregnancy.[1] Within the general population, the largest increases in obesity have been in the highest BMI groups[2] and this is also true for extreme obesity in pregnancy.[3]

Maternal obesity is a risk factor for a number of pregnancy-related complications and its relationship with these complications are complex.[4 5] These relationships can be partially explained through pre-existing comorbidities such as diabetes,[6] hypertension[6 7] and asthma.[6] Pre-existing comorbidities have been shown to increase the risk of pre-eclampsia[8 9] and venous thromboembolic events.[10] However, there remain other mechanisms that explain the association between obesity and pre-eclampsia/venous thromboembolism; to specify a few, these are inflammation,[11] insulin resistance[12] and oxidative stress.[13 14]

Several studies have investigated the prevalence, outcomes and managements of extreme obesity in pregnancy (BMI ≥50 kg/m²).[15–17] These have aimed to test whether there was a dose–response relationship between increased BMI and complications

of pregnancy. Within the extremely obese group (BMI $\geq 50\,kg/m^2$), women included have had a BMI ranging from $\geq 50\,kg/m^2$ to approximately $75\,kg/m^2$. While it may be the case that the risks rise exponentially with BMI, it is possible that above a certain BMI, the risks of maternal and perinatal complications as a result of obesity do not increase due to the competing risks of other comorbidities. This remains to be investigated, as current published data do not allow the division of women into the highest BMI groups.

Previous research pooling together international data on rare exposures in pregnancy has been limited due to heterogeneity of definitions, methods and populations.[18] The obstetric surveillance systems in Australia and the UK were designed to be compatible with data collection using similar definitions with a view to pooling data. As a result, there are comparable data available to combine national studies, which provides a large enough sample to compare two groups of women within a cohort of extremely obese women. This study aimed to compare the characteristics, management (including guideline adherence for prevention of venous thromboembolism), and maternal and perinatal outcomes of women at the extremes of obesity.

## METHODS
### Study population and design
This study was an international population-based cohort study, using secondary analysis of two national cohort studies of extreme maternal obesity, which were undertaken in Australia and the UK.[16 19] For the purposes of the analysis, the exposed cohort were those pregnant women who had a BMI $\geq 60\,kg/m^2$ and the unexposed comparison cohort were those with a BMI $>50-59.9\,kg/m^2$. Woman were included in the study if they had a BMI $>50\,kg/m^2$ at any point during the pregnancy and were included as part of the respective national studies.[16 19]

Anonymous data were prospectively collected using the respective national obstetric surveillance system, the UK Obstetric Surveillance System (UKOSS) or the Australasian Maternity Outcomes Surveillance System (AMOSS). The methods of each system have been described elsewhere in detail.[19–21] Briefly, in the UK, nominated reporters within each consultant-led obstetric unit received a monthly mailing card; the card had a tick box to indicate whether there had been a case of extreme obesity that month. There was also a box to indicate that there were no cases. Reporters returned cards regardless of whether there had been a case of extreme obesity. When a case was notified the reporter received a data collection form. Using the medical records of the patient, information on demographic characteristics, obstetric history, medical history (including height and weight), management and outcomes were collected.

A similar method was used to identify women with extreme obesity in Australia. Designated reporters within each participating maternity unit within Australia were sent a monthly email. The reporter either responded with a 'case' or a 'nil case' to indicate whether there had truly been no cases. Once a case was reported, the reporter entered data on an online data collection form using the case notes of the woman. The AMOSS system had 66% coverage of all women giving birth in Australia during the study.[19]

Outcomes, management and potential covariates relevant to the research question were identified from the literature. On the basis of this, possible covariates and outcomes were identified in the respective UKOSS and AMOSS data sets. Each variable was mapped between the AMOSS and UKOSS data sets and an assessment of the comparability was made. On occasions, where the coding differed, harmonisation of the coding was devised and applied. This resulted in uniform values and labels of variables across both data sets. An assessment was made to determine whether the variables were measuring the same clinical phenotype in similar ways.

The covariates explored in the analysis were age, smoking status during pregnancy, previous pregnancy problems, pre-existing medical problems, pre-existing hypertension, parity and multiple pregnancies.

The missing data in similar data sets have been shown not to be missing at random; as a result, multiple imputation was not considered appropriate.[22] A missing category was created for each variable to account for the missing data. Primarily, complete case analysis was used in the multivariable analysis and a sensitivity analysis including the missing categories was used to assess the impact of missing data on the point estimates.

The sample size was predetermined by the size of the existing studies; therefore, the sample was fixed at 111 women who had a BMI $\geq 60\,kg/m^2$ and 821 women who had a BMI $>50-59.9\,kg/m^2$. For the lowest frequency outcome (perinatal death), which had an incidence of 1.2% in the unexposed group, given the sample size, the minimum OR detectable as statistically significant with 80% power at the 5% significance level was 5.63. For the highest frequency outcome, which had an incidence of 66.4% (thromboprophylaxis postnatally) in the unexposed group, the minimum OR detectable as statistically significant with 80% power at the 5% significance level was 1.99.

### Statistical analysis
Descriptive analyses were undertaken using the $\chi^2$ test or Wilcoxon rank sum test as appropriate. These analyses assessed whether there was a statistical difference in characteristics between those women who had a BMI $\geq 60\,kg/m^2$ and those with a BMI $>50-59\,kg/m^2$.

Each outcome was individually modelled in a univariable analysis using unconditional logistic regression, with results presented as unadjusted OR (uOR) with 95% CI. The exposure variable in each model was extreme obesity BMI $\geq 60\,kg/m^2$. To account for clustering of infants within mothers (multiple births), robust estimates of variance were calculated. Collinearity was assessed between all plausible linear associations prior

to multivariable analysis, using Pearson's correlation coefficient.

Only outcomes that were statistically significant at the univariable level were included in the multivariable analysis. In the multivariable analysis, potential explanatory variables were sequentially added to the univariable model in a forward stepwise method with an examination of the results as each variable was added. A plausible explanatory variable was included in the final model if it was associated with the exposure and outcome (p value for Wald test <0.05) andsignificantly improved the model fit assessed by likelihood ratio tests at the 5% significance level. Statistical analysis was completed using STATA V.13.

A post hoc analysis was completed to assess the risk factors for venous thromboembolism possessed by those who did not receive postnatal thromboprophylaxis. This was a country-specific analysis using risk factors of venous thromboembolism which were identified from the Royal College of Obstetricians and Gynaecologists (RCOG) and South Australian Maternal & Neonatal Clinical Network guidelines.[23 24]

### Patient and public involvement statement

There is patient and public involvement (PPI) involvement in the UKOSS steering committee through lay members. The UKOSS steering committee assisted in the study design and management of the study. The AMOSS advisory group has PPI involvement through consumer, Maori and Pacific and Aboriginal and Torres Strait Islander members. The AMOSS advisory group provides advice on the implementation, delivery and development of the AMOSS system. The group also assists with the translation of findings into practice.

### Ethics committee approval

The Australian collaborators obtained approval for the study.[25] Ethics committee approval for secondary analysis of anonymous UK data was not required.

### RESULTS

During the period of September 2007–August 2008, 617 women with a BMI >50 kg/m$^2$ were identified through the UKOSS. Between January and October 2010, 315 women with a BMI >50 kg/m$^2$ were identified using the AMOSS. Overall, there were 111 women with a BMI ≥60 kg/m$^2$ and 821 women with a BMI >50–59.9 kg/m$^2$.

Women with a BMI ≥60 kg/m$^2$ were slightly older, the sociodemographic characteristics and previous medical histories were otherwise similar in the two groups of women (table 1).

A high proportion in both groups experienced difficulties in visualisation of ultrasound (70.3% vs 65.7%) although this was not statistically significant between the groups. Fewer women in both groups received antenatal thromboprophylaxis (24.3% BMI ≥60 kg/m$^2$ and 12.3%>50–59.9 kg/m$^2$) compared with postnatal thromboprophylaxis (77.5% and 66.4%) (table 2). Women with

a BMI ≥60 kg/m$^2$ had a significantly higher odds of pre-eclampsia/eclampsia (uOR 1.91 (95% CI 1.08 to 3.39)), and of receiving either thromboprophylaxis antenatally (uOR 2.25 (95% CI 1.39 to 3.64)) or postnatally (uOR 1.68 (95% CI 1.04 to 2.70)) compared with those with a BMI >50–59.9 kg/m$^2$ (table 2). Online supplementary tables 1 and 2 show that approximately a third of women should have received thromboprophylaxis postnatally in both the UK and Australia as they had the relevant risk factors for it to be indicated.

Although not statistically significant, a higher proportion of women with a BMI ≥60 kg/m$^2$ experienced other adverse outcomes other than pre-eclampsia/eclampsia. Pre-eclampsia/eclampsia was examined in a multivariable model. The presence of a BMI ≥60 kg/m$^2$ was associated with a twofold increase in the odds of having pre-eclampsia/eclampsia (adjusted OR (aOR) 1.83 (95% CI 1.01 to 3.30)) compared with those with a BMI >50–59 kg/m$^2$, after adjusting for smoking status, pre-existing diabetes and parity. The results of the proxy variable model did not materially differ from those of the complete case analysis.

Severe adverse outcomes such as perinatal death were uncommon in both groups (n=2 (18 per 1000 births), BMI ≥60 kg/m$^2$ vs n=10 (12 per 1000 births), BMI ≥50–59 kg/m$^2$). There were no statistically significant differences in perinatal outcomes between both obesity groups (see table 3).

### DISCUSSION
### Main findings

Compared with women with a BMI >50–59 kg/m$^2$, women with a BMI ≥60 kg/m$^2$ had an increased risk of pre-eclampsia/eclampsia, There were very few statistically significant differences in outcomes between the two very high BMI groups. Nevertheless, the direction of effects favours the lower BMI group for most outcomes. Further research should test whether any weight reduction could reduce poor outcomes even if women remain extremely obese. Importantly, the perinatal mortality rate was higher in both groups compared with the national UK and Australian rates of perinatal mortality. Women are being managed differently on the basis of BMI even at this extreme as use of thromboprophylactic drugs varied between the two high BMI groups.

### Strengths and limitations

Both prospective population-based surveillance systems use a robust methodology, which reduces the risk of selection bias. Two national studies allowed the examination of women with a BMI ≥60 kg/m$^2$ in a high resource setting and thus overcomes some of the limitations of previous research, which was limited by the number of women in the extreme ends of the BMI distribution. Nevertheless, despite pooling of national data, the number of women in each group was still relatively small, which limited

**Table 1** Sociodemographic characteristics and previous medical problems in women with body mass index (BMI) ≥60 kg/m² and comparison women (BMI >50–59.9 kg/m²)

| Characteristic | | Number (%) of obese women with BMI ≥60 kg/m² (n=111) | Number (%) of women with BMI 50–59.9 kg/m² (n=821) | P values |
|---|---|---|---|---|
| Sociodemographic characteristics | | | | |
| Age | Mean (Std) | 31.7 (5.51) | 30.3 (5.67) | 0.017 |
| BMI at booking | Median (IQR) | 61.7 (60–64.9) | 52.3 (50.8–54.9) | <0.001 |
| Max recorded BMI | Median (IQR) | 62.9 (61–66.8) | 52.7 (50.9–55.0) | <0.001 |
| Smoking status | Never/ex-smoker | 85 (76.6) | 599 (73) | 0.42 |
| | Smoked during pregnancy | 24 (21.6) | 206 (25.1) | |
| | Missing | 2 (1.8) | 16 (1.9) | |
| Known previous medical history | | | | |
| Previous pregnancy problems | None | 41 (36.9) | 273 (33.3) | 0.713 |
| | Yes | 33 (29.7) | 266 (32.4) | |
| | Primigravid | 35 (31.5) | 271 (33.0) | |
| | Missing | 2 (1.8) | 11 (1.3) | |
| Known cardiac disease | None | 109 (98.2) | 812 (98.9) | 0.2* |
| | Yes | 2 (1.8) | 5 (0.6) | |
| | Missing | 0 (0) | 4 (0.5) | |
| Known renal disease | None | 110 (99.1) | 809 (98.5) | 0.999* |
| | Yes | 1 (0.9) | 8 (1.0) | |
| | Missing | 0 (0) | 4 (0.5) | |
| Known mental health issues | None | 99 (89.2) | 756 (92.1) | 0.219 |
| | Yes | 12 (10.8) | 61 (7.4) | |
| | Missing | 0 (0) | 4 (0.5) | |
| Known asthma | None | 98 (88.3) | 720 (87.7) | 0.961 |
| | Yes | 13 (11.7) | 97 (11.8) | |
| | Missing | 0 (0) | 4 (0.5) | |
| Previous caesarean delivery | None | 53 (47.7) | 366 (44.6) | 0.297 |
| | Yes | 22 (19.8) | 181 (22.0) | |
| | Primigravid | 35 (31.5) | 271 (33.0) | |
| | Missing | 1 (0.9) | 3 (0.4) | |
| Parity | Nulliparous | 35 (31.5) | 271 (33) | 0.803 |
| | Multiparous | 75 (67.6) | 550 (67) | |
| | Missing | 1 (0.9) | 0 (0) | |
| Current pregnancy | | | | |
| Multiple pregnancy | Singleton | 108 (97.3) | 800 (97.4) | 0.928 |
| | Twin pregnancy | 3 (2.7) | 21 (2.6) | |
| Known hypertension prior to pregnancy requiring treatment | None | 103 (92.8) | 767 (93.4) | 0.693 |
| | Yes | 8 (7.2) | 51 (6.2) | |
| | Missing | 0 (0) | 3 (0.4) | |
| Known pre-existing diabetes prior to pregnancy | None | 101 (91.0) | 757 (92.2) | 0.657 |
| | Yes | 10 (9.0) | 64 (7.8) | |
| | Missing | – | – | |
| Insulin-dependent diabetes | Yes | 4 (3.6) | 17 (2.1) | 0.705* |

*Fisher's exact test

**Table 2** Maternal outcomes and management in women with body mass index (BMI) ≥60 kg/m$^2$ and comparison women (BMI >50–59.9 kg/m$^2$)

| | | Number (%) of women with BMI ≥60 kg/m$^2$ (n=111) | Number (%) of women with BMI >50–59.9 kg/m$^2$ (n=821) | Unadjusted OR | 95% CI | P values |
|---|---|---|---|---|---|---|
| **Management** | | | | | | |
| Difficulties undertaking ultrasounds | No | 30 (27) | 228 (27.8) | 1 | | |
| | Yes | 78 (70.3) | 539 (65.7) | 1.1 | (0.70 to 1.72) | 0.678 |
| | Missing | 3 (2.7) | 54 (6.6) | | | |
| Induced | No | 70 (63.1) | 505 (61.5) | 1 | | |
| | Yes | 40 (36) | 303 (36.9) | 0.95 | (0.63 to 1.44) | 0.817 |
| | Missing | 1 (0.9) | 13 (1.6) | | | |
| Caesarean delivery | No | 48 (43.2) | 398 (48.5) | 1 | | |
| | Yes | 62 (55.9) | 411 (50.1) | 1.25 | (0.84 to 1.87) | 0.274 |
| | Missing | 1 (0.9) | 12 (1.5) | | | |
| Thromboprophylaxis usage in antenatal period | No | 84 (75.7) | 706 (86) | 1 | | |
| | Yes | 27 (24.3) | 101 (12.3) | 2.25 | (1.39 to 3.64) | 0.001 |
| | Missing | 0 (0) | 14 (1.7) | | | |
| Thromboprophylaxis usage in postpartum period | No | 24 (21.6) | 255 (31.1) | 1 | | |
| | Yes | 86 (77.5) | 545 (66.4) | 1.68 | (1.04 to 2.70) | 0.033 |
| | Missing | 1 (0.9) | 21 (2.6) | | | |
| **Maternal outcome** | | | | | | |
| Wound infection in those with caesarean | No | 47 (42.3) | 344 (41.9) | 1 | | |
| | Yes | 14 (12.6) | 57 (6.9) | 1.8 | (0.93 to 3.48) | 0.081 |
| | N/A | 49 (44.1) | 410 (49.9) | | | |
| | Missing | 1 (0.9) | 10 (1.2) | | | |
| Venous thromboembolism | No | 111 (100) | 807 (98.3) | | | |
| | Yes | 0 (0) | 7 (0.9) | 0 | (0.0 to 4.00) | 0.325 |
| | Missing | 0 (0) | 7 (0.9) | | | |
| Hypertensive disorder during pregnancy | No | 77 (69.4) | 631 (76.9) | 1 | | |
| | Yes | 33 (29.7) | 183 (22.3) | 1.48 | (0.95 to 2.29) | 0.082 |
| | Missing | 1 (0.9) | 7 (0.9) | | | |
| Pregnancy-induced hypertension | No | 94 (84.7) | 702 (85.5) | 1 | | |
| | Yes | 16 (14.4) | 112 (13.6) | 1.07 | (0.61 to 1.88) | 0.823 |
| | Missing | 1 (0.9) | 7 (0.9) | | | |
| Pre-eclampsia/eclampsia | No | 93 (83.8) | 743 (90.5) | 1 | | |
| | Yes | 17 (15.3) | 71 (8.6) | 1.91 | (1.08 to 3.39) | 0.026 |
| | Missing | 1 (0.9) | 7 (0.9) | | | |

the study power, particularly when investigating rare outcomes.

This study did not have access to ethnicity from Australia and socioeconomic measures were not comparable between the countries. Thus, the aOR presented may be vulnerable to residual confounding if ethnicity and socioeconomic status were associated with both the outcome and exposure.

This analysis aimed only to compare the pregnancy outcomes of two groups of extremely obese women, and does not therefore provide any information on the outcomes of these extremely obese pregnant women in comparison to pregnant women with BMIs within the normal range. Comparisons with pregnant women who have a lower BMI have been previously published.[16 19]

**Table 3** Perinatal outcomes in women with body mass index (BMI) ≥60 kg/m² and comparison women (BMI >50–59.9 kg/m²)

| | | Number (%) of women with BMI ≥60 kg/m² | Number (%) of women with BMI >50–59.9 kg/m² | Unadjusted OR | 95% CI | P Values |
|---|---|---|---|---|---|---|
| Perinatal death* | No | 112 (98.2) | 815 (98.5) | 1 | | |
| | Yes | 2 (1.8) | 10 (1.2) | 1.46 | (0.31 to 6.74) | 0.631 |
| | Missing | 0 (0) | 2 (0.2) | | | |
| Stillbirth ≥24 weeks gestation* | No | 112 (98.2) | 818 (98.9) | 1 | | |
| | Yes | 2 (1.8) | 7 (0.8) | 2.09 | (0.43 to 10.19) | 0.363 |
| | Missing | 0 (0) | 2 (0.2) | | | |
| Preterm birth | No | 101 (90.2) | 730 (89) | 1 | | |
| | Yes | 10 (8.9) | 87 (10.6) | 0.83 | (0.36 to 1.94) | 0.668 |
| | Missing | 1 (0.9) | 3 (0.4) | | | |
| Very preterm birth | No | 111 (99.1) | 804 (98) | Omitted | | |
| | Yes | 0 (0) | 13 (1.6) | | | |
| | Missing | 1 (0.9) | 3 (0.4) | | | |
| Birth weight | Mean (Std) | 3683.0 (752.1) | 3603.7 (715.0) | Omitted | | |
| Macrosomia (>4500 g) | No | 98 (87.5) | 746 (91.0) | 1 | | |
| | Yes | 14 (12.5) | 72 (8.8) | 1.48 | (0.80 to 2.73) | 0.211 |
| | Missing | 0 (0) | 2 (0.2) | | | |
| Shoulder dystocia | No | 44 (39.3) | 373 (45.5) | 1 | | |
| | Yes | 1 (0.9) | 19 (2.3) | 0.45 | (0.06 to 3.42) | 0.438 |
| | Not applicable | 66 (58.9) | 424 (51.7) | | | |
| | Missing | 1 (0.9) | 4 (0.5) | | | |
| Congenital abnormality | No | 107 (95.5) | 797 (97.2) | 1 | | |
| | Yes | 3 (2.7) | 13 (1.6) | 1.72 | (0.48 to 6.14) | 0.404 |
| | Missing | 2 (1.8) | 10 (1.2) | | | |
| Infant respiratory problem | No | 109 (97.3) | 797 (97.2) | 1 | | |
| | Yes | 3 (2.7) | 18 (2.2) | 1.22 | (0.35 to 4.21) | 0.755 |
| | Missing | 0 (0) | 5 (0.6) | | | |
| Apgar score <7 at 5 min | No | 105 (93.8) | 778 (94.9) | 1 | | |
| | Yes | 2 (1.8) | 25 (3.0) | 0.59 | (0.14 to 2.54) | 0.482 |
| | Missing | 5 (4.5) | 17 (2.1) | | | |

OR estimated using robust standard errors.
*Denominator is birth (including multiple births) and stillbirths n=941. Denominator in the remainder of the table is live births (including multiple births).

## Interpretation

One of the novel benefits of this multinational study was the ability to examine a subset of the more extreme end of the spectrum of obesity. This demonstrated that women with a BMI ≥60 kg/m² had very similar characteristics and experienced similar management compared with women with a BMI >50 to <60 kg/m². Interestingly, BMI ≥60 kg/m² group had an increased risk of pre-eclampsia/eclampsia, which supports the hypothesis of a 'dose response' relationship between obesity and pre-eclampsia/eclampsia seen at lower BMIs[26] and super obesity.[17 27]

The comparison of extreme maternal obesity and a representative BMI group has been previously studied.[16 19] The risk of pre-eclampsia, venous thromboembolism, preterm delivery, shoulder dystocia and caesarean delivery was elevated in women with extreme maternal obesity compared with non-extremely obese women.[16 19] Despite few statistically significant differences in outcomes between the two groups, the literature highlights that the risk is substantially higher for extremely obese women compared with women in a normal BMI group.

The BMI ≥60 kg/m$^2$ cohort had a higher proportion of perinatal deaths and stillbirths than the BMI >50–59.9 kg/m$^2$ cohort, although these were not statistically significantly different possibly because of the small numbers involved. The absolute rate of perinatal death for the ≥60 kg/m$^2$ cohort was three times higher than the UK rate (5.6 per 1000 births) and 2.5 times higher than the Australian rate (7.3 per 1000); while the rate of perinatal mortality in the >50–59.9 kg/m$^2$ cohort was just over twice that of the UK rate and was 1.5 times higher than the Australian perinatal mortality rate.[28 29]

A previous study of extreme obesity that examined perinatal outcomes has suggested that there is a dose–response relationship between BMI and perinatal outcomes.[17] The small sample size and relative rarity of adverse perinatal outcomes in this analysis did not allow the role of chance to be excluded for most outcomes, even with pooling of two national studies.

The results of this study show that the degree of relative obesity impacted on thromboprophylaxis practice. The RCOG's guideline states that any women with a BMI >40 kg/m$^2$ should be considered at intermediate risk of venous thromboembolism and should be given at least 10 days of thromboprophylaxis postnatally.[23] Within Australia, there is regional variation in the guidelines concerning BMI and postpartum thromboprophylaxis. The Queensland state guideline suggests that a woman must possess three or more risk factors (BMI >30 kg/m$^2$ being one of these risk factors) to be given low molecular weight heparin for 6 days postnatally, while the South Australian government and current expert opinion recommends that BMI ≥30 kg/m$^2$ plus one major risk factor for thromboembolism requires prophylactic anticoagulation for 5 days postpartum.[24 30] Data from this study suggest guidelines appear to be followed variably due to the large variation in practice between the two cohorts. This suggests more implementation work within clinical settings is needed to help these guidelines be followed. Nevertheless, the results show that BMI has an important impact on clinical decisions concerning the administration of thromboprophylaxis postnatally.

Importantly, approximately 75% had postnatal thromboprophylaxis, which is smaller than expected considering this was an extremely obese population. Nearly a third of women in both countries had the appropriate risk factors to indicate the use of thromboprophylaxis postnatally. This highlights an important area for improvement of clinical practice to prevent a potentially fatal venous thromboembolism.

Interestingly, there were no venous thromboembolic events in the BMI ≥60 kg/m$^2$ group, which was the group in which the larger proportion of women received thromboprophylaxis, although, again, these are very rare events. Previous studies have shown that BMI is a strong risk factor of venous thromboembolism[31 32] and the risk is amplified in those who have a high BMI and were immobilised.[33]

This study was a secondary data analysis of women identified during 2008 in the UK and 2010 in the Australia. As a result, it is likely that the proportion of women who have a BMI >50 since the original studies is likely to be much larger, which makes the findings of this study even more pertinent.

## Conclusions

There were very few statistically significant differences in outcomes between these two high BMI groups. However, the direction of effect favours the lower BMI group for most outcomes and a type II error cannot be excluded given the small number of women/infants within each outcome. Pre-eclampsia risk is increased with increasing BMI in the morbidly obese women. Further research should test whether any weight reduction could reduce poor outcomes even if women remain extremely obese. Women are clearly being managed differently on the basis of BMI even at this extreme as shown by the thromboprophylaxis data. Furthermore, there was a failure to apply thromboprophylaxis guidelines fully in 2007–2008 which emphasises a need to ensure women at risk of venous thromboembolism receive appropriate prevention care.

**Acknowledgements** We thank the Australasian Maternity Outcomes Surveillance System advisory group and UK Obstetric Surveillance System steering committee for their input into the respective obesity studies. Dr Bryn Kemp and Dr Manisha Nair for their advice.

**Contributors** MK, JJK and ES conceived the study. MK, JJK, ES and SJM designed the study. ZL extracted the Australian data. SJM analysed the data and wrote the first draft. All authors interpreted the data and edited the manuscript.

**Funding** SJM is funded by the Nuffield Department of Population Health and Medical Research Council (MRC) training grant MR/K501256/1. MK is funded by an NIHR Research Professorship. The National Health and Medical Research Council Project Grant (Application 510298) funded The Australasian Maternity Outcomes Surveillance System : Improving safety and quality of maternity care in Australia (AMOSS) from 2008 to 2012.

**Disclaimer** The views expressed in this publication are those of the author(s), and not necessarily those of the MRC, the NHIR or the Department of Health. The funder had no role in the study design, data collection and analysis, decision to publish, or preparation of the article.

**Competing interests** None declared.

**Patient consent** Not required.

**Ethics approval** NSW Population and Health Services Research Ethics Committee and multiple Human Research Ethics Committees across Australia.

**Provenance and peer review** Not commissioned; externally peer reviewed.

**Data sharing statement** The NPEU Data sharing agreement can be found here: https://www.npeu.ox.ac.uk/downloads/files/npeu/policies/DataSharingPolicy.pdf. Access to the Australian data must be made to the AMOSS steering committee.

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
