## [Reviewer comments · BMJ Open]

ARTICLE DETAILS

TITLE (PROVISIONAL)	A bi-national cohort study comparing the management and outcomes of pregnancy women with a BMI>50-59.9kg/m ² and those with a BMI 60kg/m ² or greater
AUTHORS	McCall, Stephen; Li, Zhuoyang; Kurinczuk, Jenny; Sullivan, Elizabeth; Knight, Marian

VERSION 1 – REVIEW

REVIEWER	Sylvia Kirchengast University of Vienna, Austria
REVIEW RETURNED	21-Jan-2018

GENERAL COMMENTS	This is really a very interesting manuscript focusing on the increasing problem of maternal super obesity and pregnancy outcome. Unfortunately - in my opinion - the manuscript cannot be published as it stands. In detail: The main problem is the sample. The authors included exclusively morbidly obese mothers. with a BMI above 60 kg/m² and as controls mothers with a BMI between 50 .00 and 60 .00kg/m². Both groups are high risk groups and their pregnancy outcome should be compared with normal weight (BMI 18.50 to 24.99kg/m²) or overweight (BMI 25.00 to 29.99kg/m²) mothers. It makes no sense to focus on morbidly obese mothers only. There is no sufficient description how data collection took place. It is clearly not enough to write "the methods of each system are described elsewhere" All method have to be described in this manuscript. There is no description how maternal stature height and body weight have been collected. There is no information regarding age range of the participants. There is no information regarding previous births (when did they occur, problems,... There is no information regarding socioeconomic parameters. There is no information regarding ethnicity of the mothers. The sample description in table 1 is extremely poor. does multiple pregnancy means twin births? Did the authors exclude the 5 multiple births? There is no definition of including and exclusion criteria. There is no information regarding newborn size (with the exception of macrosomia) It makes no sense to include very preterm birth in the model. I recommend an extensive revision and resubmission.
--

REVIEWER	B J Hunt GSTT, UK
REVIEW RETURNED	30-Jan-2018

GENERAL COMMENTS	This paper compares outcomes in cohorts of pregnant women with BMIs between 50-59.9 and 60+ Comments  1. The introduction would benefit from a paragraph describing  a) the definition of obesity/morbid obesity and b) the mechanisms of how obesity is thought to impact on pregnancy outcome 2..Results state the period of observation was 10 years ago. Can the authors comment in the discussion that the rate of obesity is even higher now, making this data more relevant? 3. Please state in the methods all the objectives of the paper e.g The comparison between the rate of thromboprophylaxis compared to the guidelines is not mentioned in the methods, and details of the standards that the rates are being compared with would sit better there too, rather than doing this in results and discussion. 4. Please clearly define "ultrasound scanning problems"- does this mean ultrasound visualisation? 5. Thromboprophylaxis is used to reduce the rate of venous thromboembolism (VTE) not arterial events... and yet the data relating to incidence cites "thrombosis". does this include arterial thrombosis as well? Please give a breakdown of thrombotic events.AND most importantly - did the VTE occur in those who were receiving thromboprophylaxis? 6. Can the authors use other terms than > and < to describe the cohorts? Might it be easier to define the first cohort as BMI 50-59.9 and the second BMI 60 and greater? 6. 7. This review is hampered by the comparisons made of high risk groups being compared against each other. Can the results and discussion enlighten the average reader as to what rate of pre- eclampsia etc are in those with normal BMIs to give a perspective on the data? 9. The conclusion that "weight reduction of any amount prior to pregnancy could reduce poor outcomes" does not stand up for the authors have not measured the effect of weight reduction on risk. For me the conclusions are that pre-eclampsia risk is increased with increasing BMI in the morbidly obese and that there was a failure to full apply thromboprophylaxis guidelines fully in 2007-2008
---

VERSION 1 – AUTHOR RESPONSE

Reviewer: 1

Reviewer Name: Sylvia Kirchengast

Please leave your comments for the authors below This is really a very interesting manuscript focusing on the increasing problem of maternal super obesity and f pregnancy outcome..

We thank the reviewer for their interest in our paper and comments on the manuscript.

Unfortunately - in my opinion - the manuscript cannot be published as it stands.

In detail:

The main problem is the sample. The authors included exclusively morbidly obese mothers. with a BMI above 60 kg/m² and as controls mothers with a BMI between 50 .00 and 60 .00kg/m². Both groups are high risk groups and their pregnancy outcome should be compared with normal weight

(BMI 18.50 to 24.99kg/m²) or overweight (BMI 25.00 to 29.99kg/m²) mothers. It makes no sense to focus on morbidly obese mothers only.

Thank you for this comment. We would wish to clarify that the purpose of the study was to “To compare the management, maternal and perinatal outcomes of women with a BMI ≥ 60 kg/m² with women with a BMI >50 - <60 kg/m².” Thus the purpose of the study was to illustrate differences in characteristics, management and outcomes between two extremely obese cohorts not to compare to women with a normal BMI.

There is no sufficient description how data collection took place. It is clearly not enough to write "the methods of each system are described elsewhere" All method have to be described in this manuscript.

We thank the reviewer for this comment. We have now updated the methods section to include more detail about the case notification and data collection.

There is no description how maternal stature height and body weight have been collected.

Both height and weight were collected from the patient's medical records. We have updated the methods section to contain this information.

There is no information regarding age range of the participants.

We would like to refer the reviewer to table 1 where the mean age and standard deviation are presented in table 1. This gives the reader a sense of the spread of the age of the participants.

There is no information regarding previous births (when did they occur, problems,...

We thank the reviewer for this insightful comment. The data collection form for each study contained a limited amount of information on previous pregnancies. We have included the relevant information on previous pregnancies such as parity and a reported problem in a previous pregnancy and whether the woman had a previous caesarean section. We did not collect information on the dates of the previous pregnancy.

There is no information regarding socioeconomic parameters. There is no information regarding ethnicity of the mothers. The sample description in table 1 is extremely poor.

The authors acknowledge that this is a limitation to the study. Unfortunately, when combining the two studies from the UK and Australia the measures for socioeconomic status were incompatible. The data from Australia did not have ethnicity included due to a restriction imposed from an ethics committee. We have included this limitation in the discussion section

does multiple pregnancy means twin births? Did the authors exclude the 5 multiple births?

We have made it clearer that multiple pregnancy means that there were multiple infants born in the current pregnancy. The denominator changes in table 3 due to the inclusion of multiple births. We have made this clearer in table 3.

There is no definition of including and exclusion criteria.

The inclusion to the study would be women who met the case definition during the time period in either UK and Australia. We have added a statement to make this clear in the methods section.

There is no information regarding newborn size (with the exception of macrosomia) It makes no sense to include very preterm birth in the model.

Thank for you for this observation, we have added the mean birthweight and standard deviation to table 3. Each odds ratio presented in the tables is unadjusted. Each outcome (macrosomia, preterm birth etc) was modelled individually with the exposure (extreme obesity >60 vs. <60) – very preterm birth was not added to the model hence the word “omitted” beside it in the table.

I recommend an extensive revision and resubmission.

Reviewer: 2

Reviewer Name: B J Hunt

Institution and Country: GSTT, UK

Please state any competing interests or state 'None declared': None declared

Please leave your comments for the authors below This paper compares outcomes in cohorts of pregnant women with BMIs between 50-59.9 and 60+ Comments 1. The introduction would benefit from a paragraph describing

- a) the definition of obesity/morbid obesity and

We thank the reviewer for this comment and we have added the definition of obesity at the start of the introduction.

- b) the mechanisms of how obesity is thought to impact on pregnancy outcome

We thank the reviewer for this comment. We have highlighted some of key the direct and indirect mechanisms in the introduction. As our focus is not a mechanistic one, we have not expanded further in the discussion so felt this was beyond the scope of the paper.

2..Results state the period of observation was 10 years ago. Can the authors comment in the discussion that the rate of obesity is even higher now, making this data more relevant?

We thank the reviewer for this discussion point and have included this as suggested.

3. Please state in the methods all the objectives of the paper e.g The comparison between the rate of thromboprophylaxis compared to the guidelines is not mentioned in the methods, and details of the standards that the rates are being compared with would sit better there too, rather than doing this in results and discussion.

We are pleased to include this in the objective at the end of the discussion.

4. Please clearly define "ultrasound scanning problems"- does this mean ultrasound visualisation?

We apologise for the lack of clarity. We have amended the tables and text to make this clearer.

5. Thromboprophylaxis is used to reduce the rate of venous thromboembolism (VTE) not arterial events... and yet the data relating to incidence cites "thrombosis". does this include arterial thrombosis as well? Please give a breakdown of thrombotic events.AND most importantly - did the VTE occur in those who were receiving thromboprophylaxis?

We apologise that this was not clear. The thrombotic events were venous only. We have revised to 'venous thromboembolism' throughout.

There were five women who received thromboprophylaxis who had a venous thromboembolic event. We have not itemised individual events because of the risk of deductive disclosure.

6. Can the authors use other terms than > and < to describe the cohorts? Might it be easier to define the first cohort as BMI 50-59.9 and the second BMI 60 and greater? 6.

We thank the reviewer for this comment. We have amended the terms the both exposure groups throughout the document to: BMI 60 kg/m² or greater and BMI >50- 59.9kg/m².

7. This review is hampered by the comparisons made of high risk groups being compared against each other. Can the results and discussion enlighten the average reader as to what rate of pre-eclampsia etc are in those with normal BMIs to give a perspective on the data?

The comparison of extreme maternal obesity and a representative BMI group has been previously studied (Knight, 2010, Sullivan, 2015). We have highlighted this to the reader and commented on the differences in risk in the discussion section.

KNIGHT, M., KURINCZUK, J. J., SPARK, P., BROCKLEHURST, P. & SYSTEM, U. K. O. S. 2010. Extreme obesity in pregnancy in the United Kingdom. *Obstetrics & Gynecology*, 115, 989-97.

SULLIVAN, E. A., DICKINSON, J. E., VAUGHAN, G. A., PEEK, M. J., ELLWOOD, D., HOMER, C. S. E., KNIGHT, M., MCLINTOCK, C., WANG, A., POLLOCK, W., JACKSON PULVER, L., LI, Z., JAVID, N., DENNEY-WILSON, E. & CALLAWAY, L. 2015. Maternal super-obesity and perinatal outcomes in Australia: A national population-based cohort study. *BMC Pregnancy and Childbirth*, 15:322

9. The conclusion that "weight reduction of any amount prior to pregnancy could reduce poor outcomes" does not stand up for the authors have not measured the effect of weight reduction on risk. For me the conclusions are that pre-eclampsia risk is increased with increasing BMI in the morbidly obese and that there was a failure to fully apply thromboprophylaxis guidelines fully in 2007-2008

We thank the reviewer for this suggestion we have updated the conclusion according to their comments.

VERSION 2 – REVIEW

REVIEWER	Sylvia Kirchengast University of Vienna, Department of Anthropology
REVIEW RETURNED	23-Mar-2018
GENERAL COMMENTS	I am still recommending to include a normal weight control group. It makes no sense to compare pregnancy outcome of morbidly obese mothers with a BMI above 60 kg/m ² with the pregnancy outcome of obese mothers with a BMI between 50 .00 and 60 without using normal weight controls. It is nonsens to compare the management, maternal and perinatal outcomes of women with a BMI ≥60kg/m ² with women with a BMI >50-<60kg/m ² ." because both are high risk groups. I recommend to reject the manuscript

VERSION 2 – AUTHOR RESPONSE

Reviewer comments: I am still recommending to include a normal weight control group. It makes no sense to compare pregnancy outcome of morbidly obese mothers with a BMI above 60 kg/m² with the pregnancy outcome of obese mothers with a BMI between 50.00 and 60 without using normal weight controls. It is nonsense to compare the management, maternal and perinatal outcomes of women with a BMI ≥60kg/m² with women with a BMI >50-<60kg/m²." because both are high risk groups. I recommend to reject the manuscript.

Response: The specific aim of our study was to compare two groups of women, those with a BMI of 50-60kg/m² to those with a BMI of 60kg/m², to investigate whether there was any evidence of a difference in outcomes between these extremely high BMI groups. This investigation was chosen a priori, with a pre-specified protocol to this effect, to investigate whether there was evidence of a continuum of outcomes even at these very high degrees of obesity. This would inform weight management interventions for these extremely obese women and our findings do indeed indicate that there may be benefit to even small degrees of weight loss amongst this very obese cohort. This finding is of particular importance as the maternity population continues to become more obese. We are happy to provide the pre-specified study protocol on request.

Multiple studies have compared extremely obese women with women of lower or normal BMI, indeed we have previously published analyses to that effect with both of the datasets used in this research [1,

2]. We therefore believe there is little novelty in such an analysis, unlike the analysis we have conducted, which is unique.

We do, however, note the reviewer concerns and have added to the limitation section to indicate that this analysis does not provide any information comparing the outcomes of these extremely obese women to normal BMI controls. Text added as follows:

This analysis aimed only to compare the pregnancy outcomes of two groups of extremely obese women, and does not therefore provide any information on the outcomes of these extremely obese pregnant in comparison to pregnant women with BMIs within the normal range. Comparisons with pregnant women who have a lower BMI have been previously published separately (Ref Knight et al, Sullivan et al.)

References

1. Knight, M., et al., *Extreme Obesity in Pregnancy in the United Kingdom*. Obstetrics and Gynecology, 2010. **115**(5): p. 989-997.
2. Sullivan, E.A., et al., *Maternal super-obesity and perinatal outcomes in Australia: a national population-based cohort study*. BMC Pregnancy and Childbirth, 2015. **15**.